# Leptomeningeal Disease (LMD) in Patients with Melanoma Metastases

**DOI:** 10.3390/cancers15061884

**Published:** 2023-03-21

**Authors:** Mariam Lotfy Khaled, Ahmad A. Tarhini, Peter A. Forsyth, Inna Smalley, Yolanda Piña

**Affiliations:** 1Metabolism and Physiology Department, H. Lee Moffitt Cancer Center & Research Institute, Tampa, FL 33612, USA; 2Biochemistry Department, Faculty of Pharmacy, Cairo University, Cairo 12613, Egypt; 3Departments of Cutaneous Oncology and Immunology, H. Lee Moffitt Cancer Center & Research Institute, Tampa, FL 33612, USA; 4Neuro-Oncology Department, H. Lee Moffitt Cancer Center & Research Institute, Tampa, FL 33612, USA

**Keywords:** leptomeningeal disease, melanoma, cerebrospinal fluid, meninges, tumor microenvironment

## Abstract

**Simple Summary:**

Leptomeningeal disease in melanoma (LMM) patients is characterized by aggressiveness and dismal outcomes. Clinical studies and case reports demonstrate the potential of immune and targeted therapies (especially in combinatorial or multi-modal settings) in improving survival for some patients; however, LMM still progresses rapidly in most patients regardless of treatment. Several recent studies have characterized the melanoma microenvironment within the CSF compartment and improved the basic understanding of the biology of LMM. Additional laboratory and clinical studies are necessary to substantiate the relevance of different therapies and their impact on melanoma within the leptomeningeal microenvironment.

**Abstract:**

Leptomeningeal disease (LMD) is a devastating complication caused by seeding malignant cells to the cerebrospinal fluid (CSF) and the leptomeningeal membrane. LMD is diagnosed in 5–15% of patients with systemic malignancy. Management of LMD is challenging due to the biological and metabolic tumor microenvironment of LMD being largely unknown. Patients with LMD can present with a wide variety of signs and/or symptoms that could be multifocal and include headache, nausea, vomiting, diplopia, and weakness, among others. The median survival time for patients with LMD is measured in weeks and up to 3–6 months with aggressive management, and death usually occurs due to progressive neurologic dysfunction. In melanoma, LMD is associated with a suppressive immune microenvironment characterized by a high number of apoptotic and exhausted CD4^+^ T-cells, myeloid-derived suppressor cells, and a low number of CD8^+^ T-cells. Proteomics analysis revealed enrichment of complement cascade, which may disrupt the blood–CSF barrier. Clinical management of melanoma LMD consists primarily of radiation therapy, BRAF/MEK inhibitors as targeted therapy, and immunotherapy with anti-PD-1, anti-CTLA-4, and anti-LAG-3 immune checkpoint inhibitors. This review summarizes the biology and anatomic features of melanoma LMD, as well as the current therapeutic approaches.

## 1. Introduction

Leptomeningeal disease (LMD, also known as leptomeningeal carcinomatosis or carcinomatous meningitis) is the spread of cancer to the meninges surrounding the brain and the spinal cord [1]. LMD is a rare and devastating complication seen in melanoma and other advanced cancers. Patients with LMD usually have a grave prognosis, with a median survival rate ranging from weeks to a few months [2]. LMD is a challenging neuro-oncological disease for scientific researchers and clinicians as the diagnosis can be complex and effective treatments are lacking.

Malignant melanoma is the third most common solid tumor that metastasizes to the central nervous system (CNS) after lung and breast cancer [3]. Advanced melanoma is the most lethal type of skin cancer, resulting from malignantly transformed melanocytes with a high metastatic potential to migrate to different tissues, including the leptomeninges. Moreover, primary CNS melanoma can occur at the leptomeninges, which a rare but aggressive subtype of melanoma [4]. LMD is usually reported in 5% of patients with melanoma, 3–5% of patients with non-small cell lung cancer, and 10–25% of patients with breast cancer [5,6,7]. In particular, LMD from melanoma (LMM) portends an abysmal prognosis, with a mean overall survival (OS) of 3.5 months, even with aggressive treatment [8]. Retrospective studies have reported concomitant brain metastasis in 60–85% of LMM cases [9,10,11,12].

Melanoma is the 5th most common type of cancer in the United States and the 19th worldwide [13,14]. In 2020, the WHO reported 324,635 new melanoma cases and 57,043 deaths from this disease worldwide. The melanoma burden is estimated to increase to 510,000 new total cases and 96,000 deaths by 2040. Thus, the incidence of LMM is predicted to increase due to advanced treatment options for systemic disease and improved patient survival [15].

The pathogenesis of LMM is poorly understood but appears to have mechanisms similar to those of other solid cancers, including vascular spread to the leptomeninges or choroid plexus, direct extension from known brain metastases, and perineural invasion [16,17,18]. This review article focuses on the biology of the leptomeninges, how melanoma can disseminate to and survive within this space, and the immune microenvironment within the CSF of LMM patients. Additionally, we will discuss the current diagnostic and treatment modalities.

## 2. Leptomeninges and CSF Biology

Before discussing the pathogenesis of LMM, it is crucial to understand the physiological characterization of both the leptomeninges and cerebrospinal fluid (CSF).

### 2.1. Leptomeninges

Leptomeninges is the part of the meninges lining which consists of the arachnoid and pia mater, while the dura mater is referred to as the pachymeninges. The meninges (from the Greek, meninx, which means membrane) is the protective lining of the soft neural tissues of both the brain and the spinal cord. According to the meninges’ histological structure, the meninges comprise three layers (Figure 1), the dura mater (dura: tough, mater: mother), the arachnoid mater (arachne: spider), and the pia mater (pia: tender) [19]. **The dura mater** is a thick collagenous layer attached to the skull’s inner surface. It contains multiple blood vessels (mainly to supply blood to the top part of the skull) and lymphatic vessels (which act as a drain for the CSF) [19]. **The arachnoid mater** is a thin, transparent, avascular connective tissue [20]. A spongy connective meshwork located underneath the arachnoid is called arachnoid trabeculae. This meshwork comprises flattened fibroblast-like cells, scattered meningothelial cells, and rare melanocytes [21]. Openings in this meshwork constitute the subarachnoid space through which the CSF flows [22]. This subarachnoid space contains trabecular cells, collagen fibrils, arteries, veins, and the roots of cranial nerves. The arachnoid layer bridges over irregularities on the brain surface (such as gyri and sulci) and creates enlargements in the subarachnoid space, known as the subarachnoid cisterns [23]. Within the arachnoid mater, specialized projections (arachnoid villi) extend into the dural venous sinuses. The villi serve as one-way, pressure-sensitive valves that allow drainage of the CSF from the subarachnoid space into the dural venous sinuses [20]. **The pia mater** is a vascular fibro-cellular membrane that is thicker than the arachnoid and adheres to the brain surface following its contours. Blood vessels, which branch through the subarachnoid space, enter or exit the brain parenchyma with a pial sheath [23].

### 2.2. CSF

CSF is a body fluid that surrounds the brain and spinal cord and acts as a hydromechanical protector by providing physical support and buoyancy. On average, the brain weighs 1500 g; however, because of its suspension in the CSF, it only weighs 50 g [24]. Moreover, CSF maintains nourishment to the brain and spinal cord and waste removal [25]. CSF is produced mainly by specialized areas lining the brain ventricles called choroid plexuses (Figure 1). The choroid plexus is a highly specialized secretory epithelial tissue consisting of the choroid plexus epithelial cell, stroma, and fenestrated endothelial cells [26]. CSF escapes the ventricular system through the foramina of the fourth ventricle and passes into the subarachnoid space, where it is absorbed into the venous system via the arachnoid villi. Typically, the CSF production rate equals the absorption rate (0.35 mL/min or approximately 400 to 500 mL/day), which allows CSF to be renewed 4 to 5 times within 24 h. The total volume of CSF in children is 65 to 140 mL, while in adults it is 90 to 150 mL. Of the adult’s CSF volume, 25 mL is present in the ventricles and 125 mL in the subarachnoid space [26].

## 3. LMM Pathophysiology

CSF comprises 99% water, with low levels of solid constituents (e.g., cytokines, glucose, protein), which serve as a nutritional source for cell survival. Compared with blood plasma, healthy CSF contains few red blood cells or inflammatory cells [27]. Once malignant cells reach the leptomeningeal space and the CSF, they may exert different mechanisms to survive and proliferate within an almost nutrient-deprived microenvironment [28,29,30].

The invasion of the leptomeningeal layer by melanoma cancer cells can occur in different ways. The first is via direct seeding of the tumor cells to the CSF compartment during surgical resection of the brain metastasis [31]. The second is via a direct extension from the brain parenchyma, vertebral, subdural, epidural, or endoneurial/perineural metastasis [10]. The third is via hematogenous seeding via arachnoid vessels. Although LMM is commonly metastatic in origin, primary LMM occurs in rare cases, thought to arise from the resident melanocyte in the leptomeningeal space [4,32].

To characterize the immune landscape of LMM, Smalley et al. utilized single-cell RNA-Seq to profile the tumor microenvironment of LMM compared with melanoma brain metastases and skin metastases using 43 patient samples [33]. Interestingly, they found that LMM has a distinctive immune microenvironment compared with melanoma brain and skin metastasis [33]. The CSF of patients with LMM showed the highest number of apoptotic and exhausted CD4^+^ T-cells and the lowest number of CD8+ T-cells, indicating the existence of an immune-suppressed T-cell microenvironment (Figure 2) [33]. Normally, CD4^+^ and CD8^+^ T-cells provide adaptive immune T-cell immunity against tumors or foreign antigens (bacteria, viruses, and others). Depending on the signal, CD4^+^ T-cells differentiate and proliferate into different helper T-cells, which secrete cytokines to mediate the antitumor cytotoxic function of CD8^+^ T-cells [34].

The myeloid populations, including dendritic cells (DC), macrophages, monocytes, and myeloid-derived suppressor cells (MDSCs), play a diverse role in controlling tumor growth. The myeloid population in LMM specimens shows a high number of MDSC-like cells and heterogeneous populations of macrophages with markers for alternative activation [33]. The alternatively activated macrophages decrease inflammation and promote tissue repair; however, they suppress immune defense, which favors the survival of cancer cells [35]. There are some similarities in the myeloid populations between LMM and melanoma brain metastasis [33]. Tumor-associated macrophages (TAM) and MDSCs have been correlated with tumor progression and poor prognosis [36,37,38,39,40].

The immunosuppressive microenvironment has also been reported in LMD associated with breast, high-grade glioma, ovarian, esophageal, and lung cancer [41,42]. Ruan et al. investigated CSF collected from breast and lung cancer patients with LMD using single-cell RNA-Seq. Compared with control samples, patients’ CSF samples showed naïve CD4+ T-cells, exhausted and cytotoxic CD8+ T-cells, a slightly higher number of Treg, and alternatively activated macrophages [41]. Prakadan et al. profiled CSF collected longitudinally from LMD patients enrolled in two clinical trials investigating the efficiency of immune checkpoint inhibitors [42]. They found a significant increase in CD8+ lymphocytes in the CSF of patients who had received pembrolizumab (NCT02886585). They also reported an overall elevation of IFN-γ signaling and cytotoxicity in CD8 + lymphocytes post immune checkpoint inhibitors in patients enrolled in both trials. These studies emphasize the potential prevalence of the immunosuppressive microenvironment in the LMD and the efficiency of immunotherapy in modulating the immune microenvironment [33,41,42]. However, more studies with more patient samples are required to validate these findings and determine efficient immunotherapy with limited toxicity.

Specific immune subpopulations have been correlated with better clinical outcomes [33]. It was found that CSF from a good responder (who survived for more than three years on systemic targeted and immune therapy) showed a lymphocytic profile resembling the tumor-free control CSF sample. CSF from good responders had a high number of CD4^+^ and CD8^+^ T-cells and increased dendritic cells, B cells, and plasma cells. In contrast, CSF from poor responders had a high number of inactivated CD4 T-cells, MDSCs, cDC2s, and a lower number of pDCs [33].

Further studies are required to understand the reasons underpinning the functionality of the immune cells in the CSF of LMM patients. Additionally, it is essential to identify the source of these immune cells, particularly whether they are peripherally circulating immune cells that access the CSF through the disrupted blood–CSF barrier [43,44] or whether they come from the skull bone marrow, as shown in a recent bacterial meningitis study [45]. The answers to these questions could help address the mechanism of the infiltration and function of immune cells, better informing our expectation of patients’ response to therapy and disease progression.

The analysis of cellular components using scRNA-Seq of LMM CSF samples suggested an immunosuppressive microenvironment [33,41]; however, the communication mechanism between malignant melanoma and immune cells is poorly understood. Thus, investigating and analyzing the proteomic components of the CSF from LMM patients could demonstrate the mechanism of immunosuppression and melanoma survival within the leptomeninges area. A proteomic analysis of CSF samples from seven LMM patients with dismal outcomes reported 967 differentially expressed proteins compared with non-LMM controls. Pathway analysis revealed an enrichment in pathways implicated in innate immunity, including classical and alternative complement pathways, acute phase reactions, cell adhesion, platelet activation, IGF-I, GSK3 beta, and Notch signaling [46]. Moreover, the analysis revealed a high upregulation in complement cascade proteins, including C2, C3, C5, C6, C8A, and C9 in LMM patients [46]. The complement proteins provide an innate inflammatory dynamic system to host and protect the body from external antigens. The complement system could have either a pro-tumorigenic or antitumor effect in malignancies, depending on the tumor microenvironment [47,48]. In the context of leptomeningeal metastasis, the upregulation of complement component 3 (C3) has been attributed to the infiltration and survival of breast and lung cancer cells in the CSF microenvironment [30]. Boire et al. have found that C3 binds to the C3a receptor expressed on choroid plexus epithelial cells, which allows the invasion of tumor cells into the leptomeningeal area via disruption of the blood–CSF barrier [30].

The CSF environment has been shown to protect melanoma cells from MAPK-targeting therapy [46]. A BRAF-sensitive melanoma cell line (WM164) treated with BRAF inhibitor in the presence of CSF from LMM patients with a poor prognosis efficiently protected melanoma cells from apoptosis. Similarly, treating WM164 with a BRAF inhibitor in the presence of recombinant TGFβ1 (200 pg/mL) improved colony formation of the melanoma cells, despite the BRAF inhibitor, indicating the role of TGFβ1 in drug resistance. LMM patients with poor prognosis had significantly higher levels of TGFβ1 in their CSF, while CSF from LMM patients with a good prognosis and healthy control CSF did not contain any TGFβ1 [46]. It would be valuable if more studies could validate the association between the TGFβ1 level in CSF and the LMM patient therapy response. Earlier studies have reported that TGFβ1 levels in the CSF are correlated with blood–brain barrier integrity [49,50]. Screening of patients with traumatic brain injury has revealed elevated TGFβ1 in patients’ CSF, which gradually decreased over 21 days of healing. These studies unveil the role of TGFβ1 during brain damage and suggest a possible correlation between its levels and LMM patients’ prognosis [46,49,50]. However, additional studies are required to determine whether levels of TGFβ1 in CSF could be used as a relevant indicator of LMM prognosis.

## 4. Clinical Signs and Symptoms of LMD

The clinical manifestation of LMD varies depending on the location(s) of CNS involvement. Patients can present with a single neurological sign or symptom (e.g., headache, encephalopathy, seizures, gait disturbance) or multifocal signs/symptoms (Figure 3). LMD has a predilection for the cranial nerves, with diplopia, facial weakness, and hearing loss being the predominant symptoms, depending on the cranial nerve(s) involved. Normal CSF flow and volume are critical for maintaining regular intracranial pressure (ICP), which ranges between 7 and 15 mmHg in adults. ICP is controlled by intracranial blood, CSF, and neural tissue [51]. About 20–25% of patients with LMD develop CSF flow obstruction, leading to hydrocephalus and signs and symptoms of increased ICP (i.e., nausea, vomiting, headache, encephalopathy) [27]. Hydrocephalus is a condition of abnormal enlargement of the fluid-filled ventricles [52]. Hydrocephalus could be obstructive (noncommunicating) due to bulky tumor cells blocking the fourth ventricle, cerebral aqueduct, or spinal canal [53]. The other type is communicating hydrocephalus, where there is symmetrical enlargement in the ventricles, and CSF flow between ventricles is normal; however, the obstruction occurs after the CSF exits the ventricles. In some cases, increased ICP compresses the small vessels in the subarachnoid space, causing an ischemic infarction [52]. CSF spread of tumor cells can be found along the spinal axis, leading to the most frequent manifestations of limb weakness, dermatomal sensory loss, radicular pain, and bladder and/or bowel dysfunction [17]. Cerebral symptoms prevail in patients with LMD from solid malignancies, while cranial nerve impairment symptoms prevail in patients with LMD from hematological malignancies [54,55,56]; however, both solid and non-solid malignancies can have a wide range of clinical presentations.

A complete neurological examination should be carried out initially and during treatment. The Response Assessment in Neuro-Oncology-Leptomeningeal Metastases (RANO-LM) group [57,58] is developing a standardized manner to follow patients with LMD; however, no LMD-specific scorecard has been validated yet in the clinical setting.

## 5. LMD Diagnosis

LMD can be diagnosed radiographically with a concomitant clinical picture, but the “gold standard” of diagnosis remains a CSF cytology (in solid malignancies or flow cytometry in hematological malignancies) that is positive for malignancy. However, this method only relies on the morphology of the tumor cells, and lacks specific information about the molecular diagnosis. LMD can be diagnosed prior to the appearance of any clinical signs in 10–40% of cases [12,59].

### 5.1. Imaging

Contrast-enhanced cerebrospinal magnetic resonance imaging (MRI) is the approach for diagnosing and following up on the response to therapy in patients with LMD [57,60]. Complete imaging of the CNS should be obtained for staging, as LMD can be spread throughout the entire neuroaxis. The MRI should be completed prior to the lumbar puncture or shunt placement. The latter may produce pachymeningeal enhancement, confounding the results. This can give a false interpretation of findings that can be misconstrued as LMD [52]. There are many alternative diseases that show leptomeningeal enhancement on MRI, such as meningitis, encephalitis, cord compression, and multiple sclerosis [61,62,63]. Typical radiographic findings for LMD include contrast enhancement of the cranial nerves, the brain’s and lateral ventricles’ surface, cerebellar folia, basilar cisterns, cauda equina nerve roots, and lumbar nerve roots [64,65,66]. Contrast uptake can be linear, nodular, or both. By definition, the nodular areas should be at least 5 x 10 mm in size in orthogonal diameters [57]. Radiographic findings on MRI analyzed in 254 patients with LMD from solid cancers were found to be linear in 117 patients (46%), both linear and nodular in 55 patients (22%), and nodular alone in 32 patients (13%) [67]. Hydrocephalus has been reported in up to 17% of patients with LMD [68,69]. However, MRI can also be found normal in the setting of LMD [70].

### 5.2. CSF Analysis

There are two types of CSF collection specimens for medical analysis purposes (Figure 3). The first is the lumbar CSF, which is collected from the subarachnoid space in the lumbar vertebral area using a specialized needle (lumbar puncture). The second is cisternal CSF, which is collected from the cisterna magna, usually through a specialized reservoir called Ommaya (Figure 3) [71]. CSF examination is typically performed via a lumbar puncture, but CSF can also be obtained via the lumbar drain, Ommaya reservoir, or intra-operatively during surgery [72,73]. When a lumbar puncture is carried out, it is crucial to measure the opening pressure as this can guide the decision of placing a ventriculoperitoneal shunt if the ICP is elevated. CSF cytology has a high false negative rate, with about 25% to 50% of cases with LMD having positive CSF cytology with an initial lumbar puncture [55,59]. After three lumbar punctures, positive CSF cytology resulted in 90% of cases [52]. Wang et al. suggest that false-negative cytology could be avoided with three high-volume lumbar punctures and immediate processing of CSF to reduce cell death [52]. However, false-positive cytology can occur with infectious or other inflammatory diseases that involve reactive lymphocytes, as it is challenging to differentiate them from malignant cells [52,74,75]. For example, contamination of the CSF sample with peripheral blood lymphocytes from an active systemic lymphoma patient could give a false-positive cytology [76]. Additionally, the high number of mitotic B cells (CSF pleocytosis) noticed in patients with multiple sclerosis or other immune-related inflammatory diseases could also give a false-positive cytology [77,78,79,80].

Furthermore, elevated CSF pressure (>16 cmH_2_O), elevated CSF proteins (>45 mg/dL), and low CSF glucose (60 mg/dL) are signs of LMD, and are only detected in 40–70% of LMD patients [22,52]. Elevation in protein concentration has been reported in 60% to 80% of LMD patients, which could result from either blood–brain/blood–CSF barrier disruption or malignant cell secretion [54]. Elevated protein concentration can be observed in patients’ CSF collected from a lumbar puncture. On the other hand, glucose concentration goes down due to either the high metabolic turnover by the tumor cells or disrupted glucose transport [81]. However, none of these markers are specific to LMD and can be found to be abnormal in other pathological conditions.

Detection of CSF circulating tumor cells (CTCs) is a recent technique that has been tested to confirm LMD and quantify the tumor burden at the time of diagnosis and post-treatment response. CTCs with Epithelial Cell Adhesion Molecules markers (EpCAM) have been reported in patients’ blood and used for prognosis and follow-up with epithelial-based neoplasm, including breast, colorectal, lung, and prostate metastasis [82]. Thus, researchers investigated the possibility of detecting CTC in CSF to confirm LM diagnosis. In melanoma, CSF circulating melanoma cells (CSFMC) expressing CD146+, molecular weight melanoma-associated antigen (HMW-MAA) have been successfully identified using the CellSearch^®^ Veridex method [83,84,85].

Additionally, CSF circulating tumor DNA (ctDNA) has been identified in the CSF of LMM patients and could be utilized diagnostically or prognostically [86,87]. ctDNA detection in the CSF of LMD is useful as a diagnostic, prognostic biomarker and can be relevant to identify the tumor-bearing mutations at different tumor stages. Moreover, ctDNA detection can determine whether the tumor cells genetically differ from the primary tumor, which helps personalize the design of the treatment plan. Using digital droplet PCR, CSF-ctDNA analysis detected BRAF and NRAS mutations in LMM and melanoma brain metastasis patients [86,88].

Biopsy of the meninges can be considered but is rarely carried out [89,90]. A biopsy can confirm the diagnosis and exclude other conditions, such as sarcoidosis or tuberculosis, especially when primary cancers are lacking, or a patient has a history of multiple malignancies in remission.

Different assessments are recommended to enhance sensitivity in LMD diagnosis, including CSF examination and neuroimaging, along with typical clinical neurological features in a cancer patient [91,92].

### 5.3. Diagnostic Criteria for LMD

The EANO ESMO group proposed an algorithm to diagnose and categorize LMD [2]. The diagnosis is established by detecting cancer cells in the CSF (cytology for solid tumors or flow cytometry for non-solid malignancies) or biopsy (Type I, EANO ESMO classification). The CSF cytology is considered positive if tumor cells are present, equivocal if suspicious or atypical cells are present, or negative if tumor cells are absent. A second or third CSF sample should be obtained and analyzed if the initial CSF sample is negative. Suppose cytological or histological confirmation is lacking (Type II of the EANO ESMO classification) in a patient with a history of cancer. In that case, the diagnosis of LMD is probable if both typical clinical signs and MRI findings are present or possible in the presence of typical MRI findings but without the typical clinical signs. The EANO ESMO guideline was found to be highly prognostic in a retrospective cohort study [67].

## 6. LMM Prognosis

The prognosis of patients with LMD remains very poor, with a median survival of months after aggressive management. Introducing immunotherapy and targeted therapy to patients with LMD melanoma showed improvement in overall survival (OS) and neurological and cognitive functions [93,94]. However, a retrospective study performed on melanoma patients with LMD between 2011 to 2019 reported a median OS of 2.9 months among patients treated with either BRAF/MEK inhibitors or anti-PD1 [11]. More clinical studies are required to evaluate the new therapeutic modalities’ efficiency in treating melanoma patients with LMD.

## 7. LMM Treatment

EANO-ESMO [2] and NCCN [95] groups offer expert opinion consensus guidelines for clinical practice based on the limited clinical trial data on hand in LMD. Very few prospective clinical studies have been completed on patients with LMD. In addition, the interpretation of the available findings is convoluted with heterogeneity and wide-ranging response criteria, as well as having prior studies combining multiple solid tumors.

Often, palliative treatment is pursued, given the poor prognosis of patients with LMD. Options for treatment include intrathecal (IT) chemotherapy/immunotherapy, systemic targeted/immunotherapies, and/or radiation therapy. Clinical trials for LMD treatment depend on combination regimens of radiotherapy and intrathecal; however, studies have shown variable/limited response to treatment, which is limited by the toxicity [96,97].

### 7.1. Radiation Therapy

Whole brain radiation therapy (WBRT) +/− focal radiation therapy (RT) to the spine to symptomatic areas is the most common approach) [98]. WBRT has shown improved survival in LMD patients with primary lung cancer [99,100]. In some LMM patients, WBRT combined with systemic therapy has shown improved survival, potentially due to managing patients’ neurological symptoms [101,102]. However, WBRT as sole therapy did not significantly impact LMM patients [8,11]. Focal radiation treats symptomatic and nodular areas and mitigates CSF blockage areas in patients [2]. RT is a beneficial palliative treatment, especially in patients suffering from pain and/or obstructive hydrocephalus. Photon craniospinal irradiation (CSI) in LMD from solid malignancies has unfavorable side effects (i.e., bone marrow suppression and gastrointestinal toxicities) [103]. Proton CSI has less exit radiation, sparing anterior structures to the spinal canal. A phase 1 study of hypofractionated proton CSI in 24 patients with LMD from solid tumors resulted in a lower toxicity rate than historical photon CSI [103]. Four patients (19%) achieved CNS control for more than 12 months [103]. A recent phase II clinical trial reported that proton CSI significantly surpasses photon CSI in progression-free and overall survival (OS). The primary cohort recruited with LMD was mainly with breast and NSCLC, but six patients with LMM were also included (NCT04343573) [104]. Nevertheless, the restricted availability of proton CSI impedes its widespread application.

### 7.2. Systemic Therapy

Another option to treat LMD is systemic therapy, which can help control other disease sites and potentially prolong patient survival [105]. Depending on the driver mutation, cutaneous melanoma is classified into four genomic subtypes: BRAF-mutant, *RAS*-mutant, *NF*-loss, and triple wild-type [106]. Alteration in BRAF, NRAS, and NF1 causes activation in MAPK pathway, which activates uncontrolled cell proliferation [107]. Almost 60% of cutaneous melanoma patients have a BRAF driver mutation (most commonly V600E), which is targetable with an inhibitor of BRAF [108]. The first BRAF inhibitor approved by the FDA in 2011 (vemurafenib) significantly improved patient outcomes [109,110]. However, melanoma frequently develops drug resistance through the re-activation of MEK [111]. Thus, combining a MEK inhibitor with an inhibitor of BRAF is now the standard of care targeted therapy for BRAF-mutated metastatic melanoma [112,113,114]. Sakji-Dupré et al. have shown in a small cohort that there is a low penetration rate of BRAF inhibitor vemurafenib to the CSF, with a highly variable rate between patients [115]. However, some case reports have described improvements in LMM with BRAF/MEK inhibitor therapy [116,117] or in combination with RT [102,118,119].

During the last decade, there have been tremendous advancements in immunotherapy for treating late-stage cancer patients, particularly melanoma, renal cell carcinoma, leukemia, and lymphoma [120,121,122]. Immune checkpoint inhibitors (ICI) have revolutionized the era of cancer immunotherapy [123,124]. ICIs target regulatory pathways on T lymphocytes, enhancing the anti-tumor immune response [123]. ICI, including anti-PD-1 (nivolumab), anti-CTLA-4 (Ipilimumab), and anti-LAG-3 (relatlimab), are now FDA-approved for the treatment of unresectable or metastatic melanoma [125,126,127]. Unfortunately, many clinical trials have excluded patients with LMM due to their poor prognosis [127,128,129,130]. There is a critical need to develop trials that are more inclusive of patients with LMM.

Promising results from utilizing immunotherapy in metastatic melanoma treatment have encouraged researchers to explore the efficiency of immunotherapy for LMM patients [8,12,131,132]. Few retrospective studies have been conducted to evaluate immunotherapy’s efficacy on LMM patient outcomes due to the novelty of FDA approval for metastatic melanoma immunotherapy [8,12,131,132]. Interestingly, Tétu et al. have reported improved survival of five LMM patients with concomitant brain metastasis (out of 27 LMM patients) who received anti-PD-1 +/− CTLA-4 combined with radiotherapy. Those five patients had a median follow-up of 47.4 months with a complete persistent response. The remaining population’s median OS was 5.1 months [12]. Tétu et al.’s study included a total of 27 LMM patients with systemic therapy, 17 patients with immunotherapy, five patients with targeted therapy, one with chemotherapy, and four with combined therapy of anti-PD1 and a BRAF inhibitor [12]. Radiotherapy was combined with systemic treatments in nine LMM patients (31%), which included either stereotactic radiosurgery (7 patients) or WBRT (2 patients) [12].

Furthermore, an earlier study was performed on 178 LMM patients who received at least one treatment for LMM, either via radiation, systemic (chemotherapy, targeted therapy, immunotherapy), or intrathecal therapy [8]. The median OS for patients who received immunotherapy was 2.9 months, targeted therapy was 8.2 months, and radiation therapy was 4.6 months [8]. In addition, several case reports of improved survival with immunotherapy in LMM patients have been reported. A recent case report demonstrated six years of survival for a male patient diagnosed with LMM after stage IV BRAF-positive cutaneous melanoma [94]. The patient had received targeted therapy and immunotherapy systematically, in addition to intrathecal methotrexate [94]. A similar study has reported more than two years of LMM remission in a BRAF-positive melanoma patient treated with anti-PD-1 immunotherapy and radiotherapy [93]. These reports shed light on the advanced multidisciplinary treatment regimens that may improve patient outcomes and prolong their cancer-free survival. Nevertheless, ICI has been associated with various immune-related adverse effects such as arthritis, aseptic meningitis, encephalitis, and myelitis [133,134,135]. However, some of these adverse effects were manageable with corticosteroids [134,135].

Further studies are needed to uncover the factors impacting the variable response among LMM patients who received immunotherapy.

### 7.3. Intrathecal (IT) Therapy

LMD patients with normal CSF flow, as demonstrated in their neuroimaging, may benefit from IT chemotherapy as the treatment is equally distributed within the CSF compartment. However, patients with obstructed CSF flow will have the worst prognosis, and IT therapy may be ineffective, with an increased risk of developing neurotoxicity as a higher dose of the medication will be pooled proximally to the flow obstruction site [66]. Neurotoxicity from IT therapy could result from various factors, including the therapy’s type, dose, and administration route. It has been reported that frequent administration of IT via lumbar puncture can cause cord or nerve damage [136]. Furthermore, patients receiving IT therapy via the Ommaya reservoir may be subjected to aseptic or chemical meningitis (*Staphylococcus epidermidis*), leukoencephalopathy, seizures, and myelopathy [136,137].

Methotrexate, arabinofuranosyl cytidine (Ara-C), liposomal Ara-C (DepoCyt), and thiotepa are the main reported controlled trials for IT therapy. Ara-C is an antimetabolic agent which is activated intracellularly to cytarabine-5′-triphosphate. It is then incorporated into DNA during DNA synthesis, resulting in cell cycle arrest [138]. To reduce cytidine neurotoxicity, liposomal Ara-C (DepoCyt) has been alternatively used to treat leptomeningeal metastasis [139]. DepoCyt is developed through the encapsulation of cytarabine in biodegradable nano-lipid-based particles, with a gradual drug release with extended exposure in the CSF [139]. IT administration of dexamethasone with DepoCyt is necessary to reduce neurotoxic side effects. A clinical study used triple intrathecal therapy Methotrexate (12.5 mg), Cytarabine (50 mg then reduced to 15 or 25 mg) + Prednisolone (40 mg) as a prophylactic approach against leptomeningeal metastasis in patients with acute lymphoblastic leukemia [140].

In addition to chemotherapy, IT administration of immunotherapies such as IL2 and anti-PD-1 has been adopted in LMM treatment. A retrospective study on 43 patients with LMM has reported that IT administration of interleukin-2 (IL-2, important cytokines for T lymphocyte proliferation) extended the survival of these patients (median OS was 7.8 months) [141]. The range for patient survival was between 0.4 to 90.8 months which is a vast range, raising many questions regarding the patients’ tumor-immune microenvironment. This study has reported some toxicities associated with the IT administration of IL-2, including fever, chills, and symptoms of high intracranial pressure (headache and nausea) [141]. Currently, a dose escalation trial is ongoing, studying the concurrent administration of anti-PD-1 immunotherapy (nivolumab) intrathecally with different doses on day one, and intravenously (240 mg) on day 2 (NCT03025256) [132]. The initial results from this study showed that IT administration of 20 mg nivolumab was a safe and efficient dose in LMM patient’s treatment. This study collects CSF from LMM patients to measure CSF pharmacokinetics. CSF pharmacokinetics after either IT or systemic therapy is highly understudied in Leptomeningeal disease. In addition to CSF pharmacodynamics, studying the CSF immune profile of patients under immunotherapy is crucial for evaluating the immune system’s response and immunotherapy’s efficacy. A significant alteration in the CSF immune profile of LMM patients receiving targeted or immunotherapy was previously shown in [33]. The safety and efficiency of IT administration of ICI has also been reported in patients with relapsed primary CNS, encouraging the potential therapeutic impact of ICI intrathecally [142]. On the other hand, immune-mediated colitis has been reported in one patient during IT nivolumab to treat metastatic brain melanoma. However, it was unclear whether the colitis resulted from tocilizumab (anti-IL-6) administration or IT nivolumab [135].

Overall, there is an urgent need to collect CSF from cancer patients who have the propensity to metastasize to the CNS region before and after systemic therapy (either targeted or immunotherapy) to evaluate its pharmacokinetics within the CNS. Furthermore, there is a great need for additional preclinical studies to identify superior therapeutic approaches for LMM patients. A preclinical in vivo patient-derived xenograft model for LMM has been generated by injecting patients’ CTC intrathecally in mice, which could be a relevant model to assess the novel therapeutic targets’ effectiveness [143]. A murine Ommaya reservoir has been developed and could be leveraged for studying novel intrathecal therapies [143].

### 7.4. Response to Treatment

The ESMO-EANO LMD guidelines categorize eight LMD subtypes to guide treatment: I or II based on CSF cytology and A/B/C/D based on imaging features, with type (A) with linear leptomeningeal contrast enhancement only, type (B) with leptomeningeal nodules only, type (C) with both linear and nodular enhancement, and type (D) with normal MRI or hydrocephalus [2]. The Response Assessment in Neuro-Oncology (RANO) group has also provided guidelines to measure the response of patients with LMD to the treatment [57]. They reported three essential elements for RANO: neurological features examination, presence or absence of CSF circulating tumor cells (CSF cytology/flow cytometry in hematological malignancies), and neuroaxis imaging (MRI).

### 7.5. Palliative and Supportive Care

Managing and treating LMM is challenging, and multiple factors (age, tumor stage, previous therapies) can impact the treatment plan. Despite recent case reports stating that combined therapeutic approaches succeed in prolonging remission in LMM patients [12,93,94], several cases with relapsed LMD may require urgent surgical intervention and medications to alleviate neurological deficits and improve quality of life.

One of the medical emergencies that develops in most LMD cases is the elevation of intracranial pressure (ICP). Elevated ICP has no pathognomonic radiologic signs, and the most accurate way to measure or monitor ICP is by using an intraventricular catheter (Ommaya) [72]. Ommaya allows CSF drainage, resolving the elevated ICP; it is therefore a lifesaving procedure [144]. Moreover, Ommaya placement is used to deliver intrathecal treatment to patients [72]. Lumbar puncture is another procedure to relieve symptoms from hydrocephalus as well as allowing CSF collection for further analysis [52,55,73]. In cases of obstructive hydrocephalus, due to bulky tumor mass, palliative radiotherapy is an initial treatment option; however, sometimes RT cannot manage the patient’s symptoms, and CSF diversion using ventriculoperitoneal shunting is an alternative treatment [145,146,147,148].

Pain mitigation with analgesics such as nonsteroidal anti-inflammatory drugs and opioids is critical primary supportive care offered to patients. Steroid medications can alleviate headaches and radicular pain. However, as immunosuppressive agents, steroids are not recommended to be administered with immunotherapy [8]. Patients with neuropathic pain are treated with neuropathic agents, such as duloxetine and gabapentin [149]. Antiepileptic medication is indicated only in patients who experience seizures [150].

Frequent follow-up and response evaluation are essential in melanoma patients with LMD due to rapid deterioration in neurological functions and overall poor prognosis reported with those patients [8,11]. Accordingly, treatment and supportive care goals must be reassessed to avoid further deterioration.

## Figures and Tables

**Figure 1 cancers-15-01884-f001:**
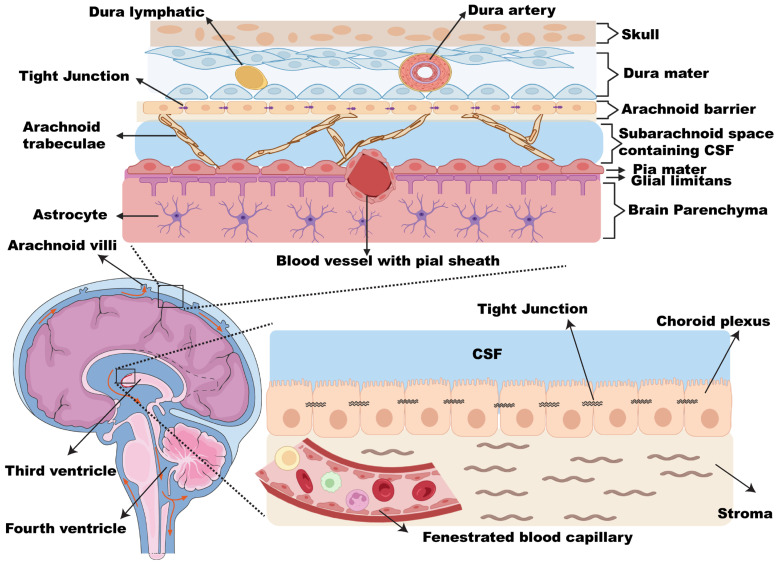
The leptomeninges and CSF biology. The side view of the brain demonstrates the third and fourth ventricles where the cerebrospinal fluid (CSF) flows (red arrows). The upper square shows the three layers of the meninges covering the brain. The lower square represents the blood–CSF barrier and shows the choroid plexus layer, the specialized lining of the brain ventricles.

**Figure 2 cancers-15-01884-f002:**
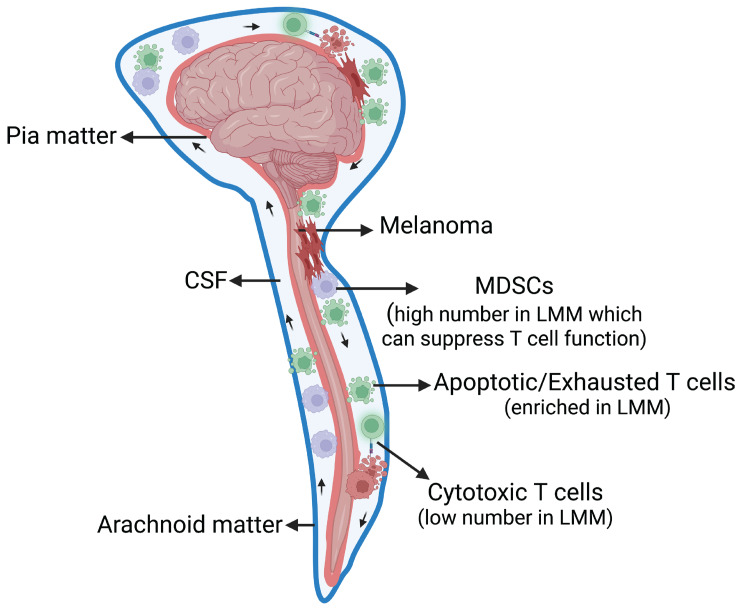
Tumor microenvironment of LMM. LMM CSF comprised a high number of apoptotic and exhausted CD4+ T-cells and a low number of CD8+ T-cells in addition to enrichment of MDSCs-like cells, indicating an immune-suppressed microenvironment ([33]).

**Figure 3 cancers-15-01884-f003:**
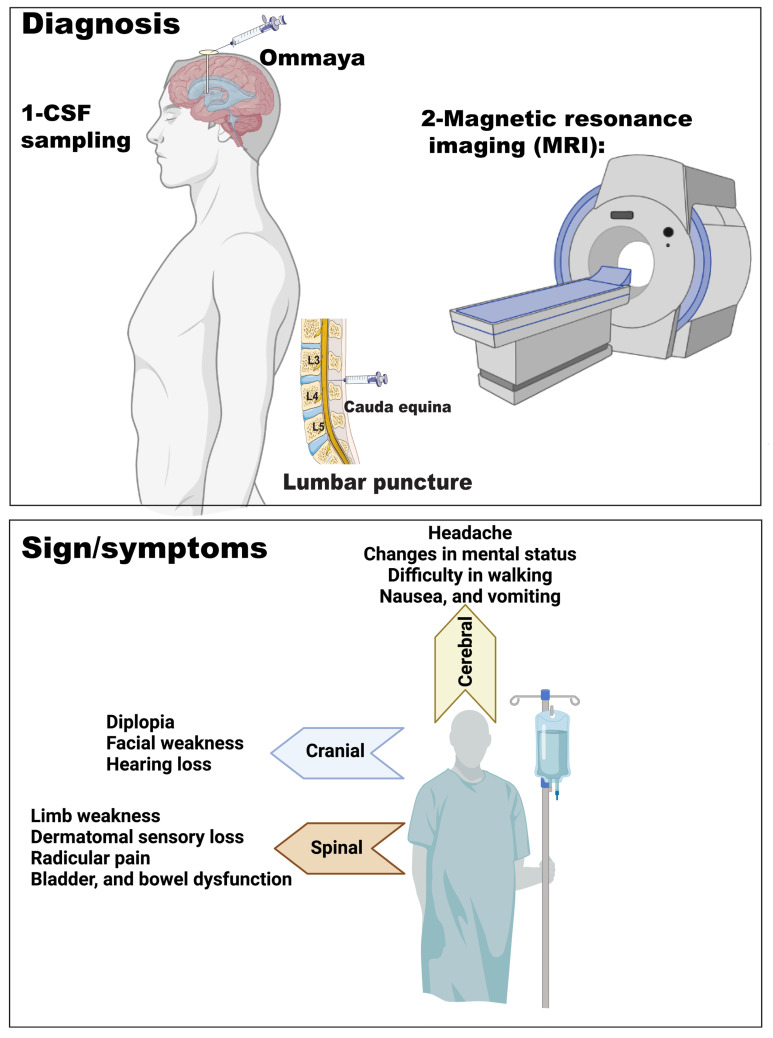
Diagnosing LMD (**upper** panel) using CSF sampling and MRI. Signs/symptoms in LMD (**lower** panel) depending on the location(s) of CNS involvement.

## Data Availability

The data can be shared up on request.

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
