# Peer review of "Leptomeningeal Disease (LMD) in Patients with Melanoma Metastases"

_cancers, 2023, doi:10.3390/cancers15061884_

Round 1
Reviewer 1 Report
Authors
Lotfy Khaled et al provide a comprehensive review of leptomeningeal disease in melanoma.
The topic is of utmost interest as LMD remains a therapeutic challenge in melanoma.
The article is well written and easy to read.
I have some minor comments and suggestions:
- Introduction L39: “melanoma is the most lethal skin cancer”: I suggest that Advanced melanoma is the most lethal skin cancer. Otherwise I think Merkel cell carcinoma is the deadliest
- Introduction L42-45: what is the percent of LMD diagnosed with or without concomitant brain metastases?
- The paragraph “4.Prognosis” could be moved before the treatment.
- 5. Clinical signs: what is the proportion of asymptomatic LMD? LMD may be diagnosed on MRI before the occurrence of any symptoms.
- The differential diagnoses of LMD could be discussed, in particular immune-related meningitis in patients treated with immune checkpoint inhibitors.
- L284: could immune-related neurological disorders be a cause of false positive cytology?
- L305-310: BRAF can also be detected in the CSF in BRAF-mutated melanoma patients, this information should be added
- L339-343: ICI and targeted therapies should be mentioned in the therapeutic options (L340) as some data were recently published in LMD and clinical trials are ongoing with these treatments, which are probably more promising than chemotherapeutic agents in melanoma.
- L346: “WBRT has shown improved survival”: I believe it is not true in LMD in melanoma. The authors should specify the solid tumors they are referring to, or update the references.
- L370: “Thus combining a MEKi with a BRAFi is now the standard of care for metastatic melanoma”: please rephrase as ipi+nivo is the standard of care; BRAFi and MEKi are the recommended targeted therapies in BRAF-mutated melanoma.
- L392: “Tetu et al have reported improved…. Combined with radiotherapy”: was it WBRT? Or stereotaxic radiosurgery ? on brain mets or LMD? Please specify.
- L436-452: was there any specific neurologic toxicities reported with the use of IT immunotherapies?
Author Response
Reviewer 1:” Lotfy Khaled et al provide a comprehensive review of leptomeningeal disease in melanoma. The topic is of utmost interest as LMD remains a therapeutic challenge in melanoma.
The article is well-written and easy to read.
Response: Thank you for reviewing our paper and for all your valuable suggestions and comments. Please consider that the line numbers have changed due to the updates.
- Introduction L39: “melanoma is the most lethal skin cancer”: I suggest that Advanced melanoma is the most lethal skin cancer. Otherwise, I think Merkel cell carcinoma is the deadliest.
Response: we agree, and we changed it to advanced melanoma (L39)
- Introduction L42-45: what is the percent of LMD diagnosed with or without concomitant brain metastases?
Response: we have added this information [lines L48-49 ]“Retrospective studies have reported concomitant melanoma brain metastasis in 60-85% of LMM cases [9-12]."
- The paragraph “4. Prognosis” could be moved before the treatment.
Response: Good suggestion. We moved the prognosis before the treatment (section 6)
- Clinical signs: what is the proportion of asymptomatic LMD? LMD may be diagnosed on MRI before the occurrence of any symptoms.
Response: we have added this piece of information under the diagnosis part [line 283] “LMD can be diagnosed prior to the appearance of any clinical signs in 10-40% of the cases [12,60]“
- The differential diagnoses of LMD could be discussed, in particular, immune-related meningitis in patients treated with immune checkpoint inhibitors.
Response: We have added this information to the LMD diagnosis [lines 291-292] “There are many alternative diseases that show leptomeningeal enhancement on MRI such as meningitis, encephalitis, cord compression, and multiple sclerosis [62,64]"
Also, we have added to the systemic therapy section this part [lines 485-488] ” Nevertheless, ICI has been associated with various immune-related adverse effects such as arthritis, aseptic meningitis, encephalitis, and myelitis [139-141]. However, some of these adverse effects were manageable with corticosteroids [140-141]"
L284: could immune-related neurological disorders be a cause of false positive cytology?
Response: yes, it is possible, that is why we mentioned on (L358) that different assessment is important. Additionally, we mentioned that after 3 lumbar punctures, sensitivity of CSF cytology reaches 90% [line 318]
We have added another paragraph to the CSF analysis section[Lines 321-327] “On the other hand, false-positive cytology can occur with infectious or other inflammatory diseases which involve reactive lymphocytes, as it is challenging to differentiate them from malignant cells [52, 76, 77]. For example, contamination of the CSF sample with peripheral blood lymphocytes from an active systemic lymphoma patient could give a false positive cytology [78]. Additionally, the high number of mitotic B cells (CSF pleocytosis) noticed in patients with multiple sclerosis or other immune-related inflammatory diseases [79-82]"
- L305-310: BRAF can also be detected in the CSF in BRAF-mutated melanoma patients; this information should be added
Response: This valuable information has been added to the review [line 351] “Using digital droplet PCR, CSF-ctDNA analysis detected BRAF and NRAS mutations in LMM and melanoma brain metastasis patients [90,91]."
- L339-343: ICI and targeted therapies should be mentioned in the therapeutic options (L340) as some data were recently published in LMD, and clinical trials are ongoing with these treatments, which are probably more promising than chemotherapeutic agents in melanoma.
Response: we have updated this section [line 405] “Often, palliative treatment is pursued, given the poor prognosis of patients with LMD. Options for treatment include intrathecal (IT) chemotherapy/immunotherapy, systemic targeted/ immunotherapies, and/or radiation therapy.”
- L346: “WBRT has shown improved survival”: I believe it is not true in LMD in melanoma. The authors should specify the solid tumors they are referring to or update the references.
Response: we agree with the reviewer, and we have updated this section [lines 412-416] ” WBRT has shown improved survival in LMD patients with primary lung cancer [103, 104]. In some LMM patients, WBRT combined with systemic therapy has shown improved survival potentially due to managing patients' neurological symptoms [105, 106]. However, WBRT as sole therapy did not significantly impact LMM patients [8, 11]."
- L370: “Thus combining a MEKi with a BRAFi is now the standard of care for metastatic melanoma”: please rephrase as ipi+nivo is the standard of care; BRAFi and MEKi are the recommended targeted therapies in BRAF-mutated melanoma.
Response: thank you for raising this critical correction, we have corrected it [line 444]. “Thus, combining a MEK inhibitor with an inhibitor of BRAF is now the standard of care targeted therapy for BRAF-mutated metastatic melanoma.”
- L392: “Tetu et al. have reported improved…. Combined with radiotherapy”: was it WBRT? Or stereotaxic radiosurgery? On brain mets or LMD? Please specify.
Response: We have clarified it [lines 463-472] “Tétu et al. have reported improved survival of five LMM patients with concomitant brain metastasis (out of 27 LMM patients) who received anti-PD-1 +/- CTLA-4 combined with radiotherapy. Those five patients had a median follow-up of 47.4 months with a complete persistent response. The remaining population's median OS was 5.1 months [12]. Tétu et al. study included a total of 27 LMM patients with systemic therapy, 17 patients received immunotherapy, five patients received targeted therapy, one with chemotherapy, and four had combined therapy of anti-PD1 and a BRAF inhibitor [12]. Radiotherapy was combined with systemic treatments in nine LMM patients (31%), which included either stereotactic radiosurgery (7 patients) or WBRT (2 patients)[12]."
- L436-452: was there any specific neurologic toxicities reported with the use of IT immunotherapies?
Response: we have added this section [lines 527-544] “This study has reported some toxicities associated with the IT administration of IL-2 including fever, chills, and symptoms of high intracranial pressure (headache and nausea)[147]. Currently, a dose escalation trial studying the concurrent administration of anti-PD-1 immunotherapy (nivolumab) intrathecally with different doses on day one and intravenously (240 mg) on day 2 is ongoing (NCT03025256) [148]. The initial results from this study showed that IT administration of 20 mg nivolumab was safe and efficient dose in LMM patient's treatment. This study collects CSF from LMM patients to measure CSF pharmacokinetics. CSF pharmacokinetics after either IT or systemic is highly under-studied in Leptomeningeal disease. In addition to CSF pharmacodynamics, studying the CSF immune profile of patients under immunotherapy is crucial for evaluating the immune system's response and immunotherapy's efficacy. A significant alteration in the CSF immune profile of LMM patients receiving targeted or immunotherapy was previously shown in [33]. The safety and efficiency of IT administration of ICI have also been reported in patients with relapsed primary CNS encouraging the potential therapeutic impact of ICI intrathecally [149]. On the other hand, immune-mediated colitis has been reported in one patient during IT nivolumab to treat metastatic brain melanoma. However, it was unclear whether the colitis resulted from tocilizumab (anti-IL-6) administration or IT nivolumab [141]”
Reviewer 2 Report
The article is a comprehensive review of leptomeningeal disease (LMD) in patients with metastatic melanoma, including the pathophysiology of the disease, diagnostic methods, prognosis, and current treatments.
The issues included in each part of the review are complete and updated, with adequate references used.
Only a minor correction is suggested:
Line 286“…malignant lymphocytes. “ change by “…malignant cells.”
Author Response
Thank you for reviewing our review paper and for your generous comment. We agree, it is a good suggestion, and we will change it.